# Self-Gradient Compensation of Full-Tensor Airborne Gravity Gradiometer

**DOI:** 10.3390/s19081950

**Published:** 2019-04-25

**Authors:** Xuewu Qian, Yanhua Zhu

**Affiliations:** 1School of Automation and Electrical Engineering, LinYi University, Linyi 276000, China; 2School of Instrument Science and Engineering, Southeast University, Nanjing 210096, China; zhuyh@seu.edu.cn

**Keywords:** self-gradient compensation, full-tensor airborne gravity gradiometer, gravity gradiometry, rotating accelerometer gravity gradiometer

## Abstract

In the process of airborne gravity gradiometry for the full-tensor airborne gravity gradiometer (FTAGG), the attitude of the carrier and the fuel mass will seriously affect the accuracy of gravity gradiometry. A self-gradient is the gravity gradient produced by the surrounding masses, and the surrounding masses include distribution mass for the carrier mass and fuel mass. In this paper, in order to improve the accuracy of airborne gravity gradiometry, a self-gradient compensation model is proposed for FTAGG. The self-gradient compensation model is a fuction of attitude for carrier and time, and it includes parameters ralated to the distribution mass for the carrier. The influence of carrier attitude and fuel mass on the self-gradient are simulated and analyzed. Simulation shows that the self-gradient tensor element $\Gamma_{xx},\Gamma_{xy},\Gamma_{xz},\Gamma_{yz}$ and $\Gamma_{zz}$ are greatly affected by the middle part of the carrier, and the self-gradient tensor element $\Gamma_{yz}$ is affected by the carrier’s fuel mass in three attitudes. Further simulation experiments show that the presented self-gradient compensation method is valid, and the error of the self-gradient compensation is within 0.1 Eu. Furthermore, this method can provide an important reference for improving the accuracy of aviation gravity gradiometry.

## 1. Introduction

Gravity gradient measurement plays a key role in inertial navigation, mineral exploration, topographic map matching, geoscience research and many other fields [1,2,3,4]. The father of gravity gradiometry was Baron Lorand von $E\ddot{o}\mathrm{tv}\ddot{o}s$ (1848–1919), a Hungarian nobleman and a physicist and engineer. The physics unit the ‘eotvos’ or Eu (1 Eu = 0.1 mGal/km = 10${}^{-9}$ s${}^{-2}$) is now standard for characterising how sensitive different gravity gradiometers are. What was invented by Baron Lorand von $E\ddot{o}\mathrm{tv}\ddot{o}s$ was his famous ‘torsion balance gradiometer’. The instrument was used extensively for oil exploration during the early 20th century. However, it took several hours to measure a single point and required a quiet environment; therefore, its laborious setup and its high sensitivity to near field masses contributed to its disuse soon after the development of the relative gravimeter [5]. The requirement to survey larger areas in order to discern the regional variations of the gravity field was very labor intensive and often impossible to fulfill due to inaccessibility in a tough area, or the ocean, etc. [6]. In the 1960s, airborne gravimetry was developed. There have been many kinds of gravity gradiometer instruments (GGIs) or gravity gradiometers since the 1970s, such as the rotating gravity gradiometer, rotating accelerometer gravity gradiometer, Cold-Atom interferometric gravity gradiometer, superconducting gravity gradiometer and so on [7,8,9,10]. Among all of them, the rotating accelerometer gravity gradiometer is the only one that has been used on an airborne and shipborne platform and has been put into commercial operation successfully. Based on the rotating accelerometer gravity gradient measuring principle, Lockheed Martin and BHP Billiton jointly developed a part of a tensor airborne gravity gradiometer (Falcon, with 8 accelerometers), Bell Aerospace (Bedford, TX, USA) (now Lockheed Martin, Bethesda, MD, USA) developed a full tensor gradiometer (Air-FTG, with 3 rotating discs), and ARKeX Ltd has developed a full tensor gravity gradiometer (FTGeX) with the help of the Lockheed Martin. All three kinds of gravity gradiometers have carried out a great deal of energy geological exploration, and achieved a very good result [11]. Currently, the static noise density level of Gravity gradiometer Falcon, Air-FTG and FTGeX is 3 Eu/$\sqrt{Hz}$, 11 Eu/$\sqrt{Hz}$ and 7 Eu/$\sqrt{Hz}$, respectively [12]. Lee pointed out that the gravity gradiometer noise density level cannot be greater than 14 Eu/$\sqrt{Hz}$ in order to achieve energy exploration.

Development of the FTAGG is an area of active research which has already demonstrated practical utility. Use of airborne gravity gradiometry is becoming more prevalent in surveys for natural resource prospecting [13]. FTAGG is a high precision measuring instrument, which is extremely sensitive to its operating environment, and so environmental parameters such as temperature, humidity and air pressure must be strictly controlled [14]. The calibration of gravity gradient is needed before airborne gravity gradiometry can be carried out, which has published by me [15]. Another calibration method based on centrifugal gradient has been proposed [16]. This method seems novel, but it is difficult to implement, and various interference factors will affect the calibration accuracy of gravity gradiometer.

In the process of airborne gravity gradiometry, because FTAGG is extremely sensitive to its operating environment, the attitude of the carrier and the fuel mass will seriously affect the accuracy of gravity gradiometry for the FTAGG. In this paper, we propose a self-gradient compensation method for FTAGG. First, we proposed a self-gradient compensation model for FTAGG based on the relationship between the gravity gradient tensor matrix in diffent frame. Then, we simulate and analyze the self-gradient caused by the distribution mass for the carrier and fuel mass with the attitude. Finally, we design a simulation test to analyze the validity of the self-gradient compensation method.

## 2. Compensation Methods of Self-Gradient

Gravity gradiometry data are measurements of the derivatives of the components of the gravity vector $g=(g_{x},g_{y},g_{z},)$ in three orthogonal directions of space (x,y,z) [17]. The gravity gradient tensor can be denoted by
(1)$$\begin{array}{c} \begin{array}{c} \begin{array}{c} \Gamma =\left[\begin{array}{ccc} \Gamma_{xx} & \Gamma_{xy} & \Gamma_{xz}\\ \Gamma_{yx} & \Gamma_{yy} & \Gamma_{yz}\\ \Gamma_{zx} & \Gamma_{zy} & \Gamma_{zz} \end{array}\right]\;\;\;\;\;\;\;\;\;\;\;\;\;\;\;\;\\ where,\;\Gamma_{ij}=\frac{\partial g_{i}}{\partial j}=\frac{\partial g_{j}}{\partial i},\;\forall i,j\in \left\{x,y,z\right\}. \end{array} \end{array} \end{array}$$

The diagonal and off-diagonal components of Γ are called the *in-line* and cross gradient, respectively. The tensor is symmetric since the gravitational potential is a smooth function. Moreover, in free space, the Γ is traceless [6].

The basic principle of gravity gradiometry is analyzed in the instrument frame. Taking the center of the GGI disc as the origin O, establish the ENU direction as the FTAGG frame ox${}_{g}$*y*${}_{g}$*z*${}_{g}$ (g-frame). The mass distribution of the carrier is considered to be a homogeneous cuboid, establish the carrier frame ox${}_{b}$*y*${}_{b}$*z*${}_{b}$ (b-frame). In order to facilitate the description of the positional change of the carrier relative to the FTAGG, the heading angle, the pitch angle and the roll angle of the carrier attitude are ψ, λ, φ, respectively. The rotation matrices for each of the axes (*x*,*y*,*z*) are given by
$$R_{x}\left(\lambda \right)=\left[\begin{array}{ccc} 1 & 0 & 0\\ 0 & \cos\lambda & \sin\lambda \\ 0 & -\sin\lambda & \cos\lambda \end{array}\right],R_{y}\left(\varphi \right)=\left[\begin{array}{ccc} \cos\varphi & 0 & -\sin\varphi \\ 0 & 1 & 0\\ \sin\varphi & 0 & \cos\varphi \end{array}\right],R_{z}\left(\psi \right)=\left[\begin{array}{ccc} \cos\psi & \sin\psi & 0\\ -\sin\psi & \cos\psi & 0\\ 0 & 0 & 1 \end{array}\right].$$
where $R_{j}\left(\theta \right)$ represents a rotation about the *j*th axes by an angle θ. According to the rotation principle of the frame, the transformation matrix from g-frame to the b-frame can be written as:(2)$$\begin{array}{c} \begin{array}{cc} C_{g}^{b} & =R_{y}\left(\varphi \right)R_{x}\left(\lambda \right)R_{z}\left(\psi \right)\\ & =\left[\begin{array}{ccc} \cos\varphi & 0 & -\sin\varphi \\ 0 & 1 & 0\\ \sin\varphi & 0 & \cos\varphi \end{array}\right]\left[\begin{array}{ccc} 1 & 0 & 0\\ 0 & \cos\lambda & \sin\lambda \\ 0 & -\sin\lambda & \cos\lambda \end{array}\right]\left[\begin{array}{ccc} \cos\psi & \sin\psi & 0\\ -\sin\psi & \cos\psi & 0\\ 0 & 0 & 1 \end{array}\right]\\ & =\left[\begin{array}{ccc} \cos\varphi \cos\psi -\sin\varphi \sin\lambda \sin\psi & \cos\varphi \sin\psi +\sin\varphi \sin\lambda \cos\psi & -\sin\varphi \cos\lambda \\ -\cos\lambda \sin\psi & \cos\lambda \cos\psi & \sin\lambda \\ \sin\varphi \cos\psi +\cos\varphi \sin\lambda \sin\psi & \sin\varphi \sin\psi -\cos\varphi \sin\lambda \cos\psi & \cos\varphi \cos\lambda \end{array}\right]. \end{array} \end{array}$$

In the platform-type inertial navigation system, the carrier’s flight attitude can be measured by the angle sensor of the three frames of the stable platform. Since the carrier’s attitude changes, the carrier coordinate system no longer coincides with the disc coordinate system. The gravity gradient tensor in the carrier coordinate system is then multiplied by a coordinate transformation matrix on the left and right sides of the gravity gradient tensor matrix obtained in the carrier coordinate system, thereby obtaining the gravity gradient tensor in the disc coordinate system. It is assumed that the gravity gradient tensor matrix in the g-frame and b-frame are $\Gamma^{g}$ and $\Gamma^{b}$, respectively, and the conversion relationship between the gravity gradient tensor matrices under the two frames g-frame and b-frame can be obtained according to the matrix transformation relationship is: (3)$$\begin{array}{c} \begin{array}{c} \begin{array}{c} \Gamma^{g}={\left(C_{g}^{b}\right)}^{T}\Gamma^{b}C_{g}^{b} \end{array}. \end{array} \end{array}$$

The matrix $C_{g}^{b}$, $\Gamma^{g}$, $\Gamma^{b}$ are given by
$$C_{g}^{b}=\left[\begin{array}{ccc} C_{11} & C_{12} & C_{13}\\ C_{21} & C_{22} & C_{23}\\ C_{31} & C_{32} & C_{33} \end{array}\right],\Gamma^{g}=\left[\begin{array}{ccc} \Gamma_{xx}^{g} & \Gamma_{xy}^{g} & \Gamma_{xz}^{g}\\ \Gamma_{yx}^{g} & \Gamma_{yy}^{g} & \Gamma_{yz}^{g}\\ \Gamma_{zx}^{g} & \Gamma_{zy}^{g} & \Gamma_{zz}^{g} \end{array}\right],\Gamma^{b}=\left[\begin{array}{ccc} \Gamma_{xx}^{b} & \Gamma_{xy}^{b} & \Gamma_{xz}^{b}\\ \Gamma_{yx}^{b} & \Gamma_{yy}^{b} & \Gamma_{yz}^{b}\\ \Gamma_{zx}^{b} & \Gamma_{zy}^{b} & \Gamma_{zz}^{b} \end{array}\right].$$

Substituting $C_{g}^{b}$, $\Gamma^{g}$, $\Gamma^{b}$ into Equation (Equation 3), will yield the gravity gradient tensor in g-frame: (4)$$\begin{array}{c} \begin{array}{c} \begin{array}{cc} \Gamma^{g} & ={\left(C_{g}^{b}\right)}^{T}\Gamma^{b}C_{g}^{b}\\ & ={\left[\begin{array}{ccc} C_{11} & C_{12} & C_{13}\\ C_{21} & C_{22} & C_{23}\\ C_{31} & C_{32} & C_{33} \end{array}\right]}^{T}\left[\begin{array}{ccc} \Gamma_{xx}^{b} & \Gamma_{xy}^{b} & \Gamma_{xz}^{b}\\ \Gamma_{yx}^{b} & \Gamma_{yy}^{b} & \Gamma_{yz}^{b}\\ \Gamma_{zx}^{b} & \Gamma_{zy}^{b} & \Gamma_{zz}^{b} \end{array}\right]\left[\begin{array}{ccc} C_{11} & C_{12} & C_{13}\\ C_{21} & C_{22} & C_{23}\\ C_{31} & C_{32} & C_{33} \end{array}\right]=\left[\begin{array}{ccc} \Gamma_{xx}^{g} & \Gamma_{xy}^{g} & \Gamma_{xz}^{g}\\ \Gamma_{yx}^{g} & \Gamma_{yy}^{g} & \Gamma_{yz}^{g}\\ \Gamma_{zx}^{g} & \Gamma_{zy}^{g} & \Gamma_{zz}^{g} \end{array}\right]. \end{array} \end{array} \end{array}$$

Its matrix elements are given by
$$\left\{ \begin{array}{c} \Gamma_{xx}^{g}=C_{11}\left(C_{11}\Gamma_{xx}^{b}+C_{21}\Gamma_{yx}^{b}+C_{31}\Gamma_{zx}^{b}\right)+C_{21}\left(C_{11}\Gamma_{xy}^{b}+C_{21}\Gamma_{yy}^{b}+C_{31}\Gamma_{zy}^{b}\right)+C_{31}\left(C_{11}\Gamma_{xz}^{b}+C_{21}\Gamma_{yz}^{b}+C_{31}\Gamma_{zz}^{b}\right)\\ \Gamma_{xy}^{g}=C_{12}\left(C_{11}\Gamma_{xx}^{b}+C_{21}\Gamma_{yx}^{b}+C_{31}\Gamma_{zx}^{b}\right)+C_{22}\left(C_{11}\Gamma_{xy}^{b}+C_{21}\Gamma_{yy}^{b}+C_{31}\Gamma_{zy}^{b}\right)+C_{32}\left(C_{11}\Gamma_{xz}^{b}+C_{21}\Gamma_{yz}^{b}+C_{31}\Gamma_{zz}^{b}\right)\\ \Gamma_{xz}^{g}=C_{13}\left(C_{11}\Gamma_{xx}^{b}+C_{21}\Gamma_{yx}^{b}+C_{31}\Gamma_{zx}^{b}\right)+C_{23}\left(C_{11}\Gamma_{xy}^{b}+C_{21}\Gamma_{yy}^{b}+C_{31}\Gamma_{zy}^{b}\right)+C_{33}\left(C_{11}\Gamma_{xz}^{b}+C_{21}\Gamma_{yz}^{b}+C_{31}\Gamma_{zz}^{b}\right)\\ \Gamma_{yx}^{g}=C_{11}\left(C_{12}\Gamma_{xx}^{b}+C_{22}\Gamma_{yx}^{b}+C_{32}\Gamma_{zx}^{b}\right)+C_{21}\left(C_{12}\Gamma_{xy}^{b}+C_{22}\Gamma_{yy}^{b}+C_{32}\Gamma_{zy}^{b}\right)+C_{31}\left(C_{12}\Gamma_{xz}^{b}+C_{22}\Gamma_{yz}^{b}+C_{32}\Gamma_{zz}^{b}\right)\\ \Gamma_{yy}^{g}=C_{12}\left(C_{12}\Gamma_{xx}^{b}+C_{22}\Gamma_{yx}^{b}+C_{32}\Gamma_{zx}^{b}\right)+C_{22}\left(C_{12}\Gamma_{xy}^{b}+C_{22}\Gamma_{yy}^{b}+C_{32}\Gamma_{zy}^{b}\right)+C_{32}\left(C_{12}\Gamma_{xz}^{b}+C_{22}\Gamma_{yz}^{b}+C_{32}\Gamma_{zz}^{b}\right)\\ \Gamma_{yz}^{g}=C_{13}\left(C_{12}\Gamma_{xx}^{b}+C_{22}\Gamma_{yx}^{b}+C_{32}\Gamma_{zx}^{b}\right)+C_{23}\left(C_{12}\Gamma_{xy}^{b}+C_{22}\Gamma_{yy}^{b}+C_{32}\Gamma_{zy}^{b}\right)+C_{33}\left(C_{12}\Gamma_{xz}^{b}+C_{22}\Gamma_{yz}^{b}+C_{32}\Gamma_{zz}^{b}\right)\\ \Gamma_{zx}^{g}=C_{11}\left(C_{13}\Gamma_{xx}^{b}+C_{23}\Gamma_{yx}^{b}+C_{33}\Gamma_{zx}^{b}\right)+C_{21}\left(C_{13}\Gamma_{xy}^{b}+C_{23}\Gamma_{yy}^{b}+C_{33}\Gamma_{zy}^{b}\right)+C_{31}\left(C_{13}\Gamma_{xz}^{b}+C_{23}\Gamma_{yz}^{b}+C_{33}\Gamma_{zz}^{b}\right)\\ \Gamma_{zy}^{g}=C_{12}\left(C_{13}\Gamma_{xx}^{b}+C_{23}\Gamma_{yx}^{b}+C_{33}\Gamma_{zx}^{b}\right)+C_{22}\left(C_{13}\Gamma_{xy}^{b}+C_{23}\Gamma_{yy}^{b}+C_{33}\Gamma_{zy}^{b}\right)+C_{32}\left(C_{13}\Gamma_{xz}^{b}+C_{23}\Gamma_{yz}^{b}+C_{33}\Gamma_{zz}^{b}\right)\\ \Gamma_{zz}^{g}=C_{13}\left(C_{13}\Gamma_{xx}^{b}+C_{23}\Gamma_{yx}^{b}+C_{33}\Gamma_{zx}^{b}\right)+C_{23}\left(C_{13}\Gamma_{xy}^{b}+C_{23}\Gamma_{yy}^{b}+C_{33}\Gamma_{zy}^{b}\right)+C_{33}\left(C_{13}\Gamma_{xz}^{b}+C_{23}\Gamma_{yz}^{b}+C_{33}\Gamma_{zz}^{b}\right) \end{array}\right.$$

In order to conveniently represent the gravity gradient tensor element, we introduce a gravity gradient vector and a gravity gradient vector matrix to describe the transformation of the gravity gradient among different frames. The gravity gradient tensor can be converted into a gravity gradient element vector form, which is expressed as follows: (5)$$\begin{array}{c} \begin{array}{c} \begin{array}{c} T^{g}=\left(A_{b}^{g}\right)T^{b} \end{array}. \end{array} \end{array}$$
where $A_{b}^{g}$ denotes the matrix of the gravity gradient tensor, $T^{g}$ and $T^{b}$ denote the elements of the gravity gradient tensor in g-frame and in b-frame, respectively. The gravity gradient tensor element vector in g-frame and in b-frame can be defined as: (6)$$\begin{array}{c} \begin{array}{c} \begin{array}{c} \left\{\begin{array}{c} T^{g}={\left[\begin{array}{cccccc} \Gamma_{xx}^{g} & \Gamma_{xy}^{g} & \Gamma_{xz}^{g} & \Gamma_{yy}^{g} & \Gamma_{yz}^{g} & \Gamma_{zz}^{g} \end{array}\right]}^{T}\\ T^{b}={\left[\begin{array}{cccccc} \Gamma_{xx}^{b} & \Gamma_{xy}^{b} & \Gamma_{xz}^{b} & \Gamma_{yy}^{b} & \Gamma_{yz}^{b} & \Gamma_{zz}^{b} \end{array}\right]}^{T} \end{array}.\right. \end{array} \end{array} \end{array}$$

The element of the gravity gradient tensor vector $T^{g}$ and $T^{b}$ are the upper triangular matrix of the gravity gradient tensor in g-frame and b-frame, respectively. The gravity gradient vector matrix $A_{b}^{g}$ is obtained by rearranging the elements of the matrix $\Gamma^{g}$ and extracting the constants without gravity gradient elements. So, from Equations (Equation 4) and (Equation 5), we obtain the relational expression of the gravity gradient tensor vector: (7)$$\begin{array}{cc} \left[\begin{array}{c} \Gamma_{xx}^{g}\\ \Gamma_{xy}^{g}\\ \Gamma_{xz}^{g}\\ \Gamma_{yy}^{g}\\ \Gamma_{yz}^{g}\\ \Gamma_{zz}^{g} \end{array}\right] & =\left[\begin{array}{c} C_{11}\left(C_{11}\Gamma_{xx}^{b}+C_{21}\Gamma_{yx}^{b}+C_{31}\Gamma_{zx}^{b}\right)+C_{21}\left(C_{11}\Gamma_{xy}^{b}+C_{21}\Gamma_{yy}^{b}+C_{31}\Gamma_{zy}^{b}\right)+C_{31}\left(C_{11}\Gamma_{xz}^{b}+C_{21}\Gamma_{yz}^{b}+C_{31}\Gamma_{zz}^{b}\right)\\ C_{12}\left(C_{11}\Gamma_{xx}^{b}+C_{21}\Gamma_{yx}^{b}+C_{31}\Gamma_{zx}^{b}\right)+C_{22}\left(C_{11}\Gamma_{xy}^{b}+C_{21}\Gamma_{yy}^{b}+C_{31}\Gamma_{zy}^{b}\right)+C_{32}\left(C_{11}\Gamma_{xz}^{b}+C_{21}\Gamma_{yz}^{b}+C_{31}\Gamma_{zz}^{b}\right)\\ C_{13}\left(C_{11}\Gamma_{xx}^{b}+C_{21}\Gamma_{yx}^{b}+C_{31}\Gamma_{zx}^{b}\right)+C_{23}\left(C_{11}\Gamma_{xy}^{b}+C_{21}\Gamma_{yy}^{b}+C_{31}\Gamma_{zy}^{b}\right)+C_{33}\left(C_{11}\Gamma_{xz}^{b}+C_{21}\Gamma_{yz}^{b}+C_{31}\Gamma_{zz}^{b}\right)\\ C_{12}\left(C_{12}\Gamma_{xx}^{b}+C_{22}\Gamma_{yx}^{b}+C_{32}\Gamma_{zx}^{b}\right)+C_{22}\left(C_{12}\Gamma_{xy}^{b}+C_{22}\Gamma_{yy}^{b}+C_{32}\Gamma_{zy}^{b}\right)+C_{32}\left(C_{12}\Gamma_{xz}^{b}+C_{22}\Gamma_{yz}^{b}+C_{32}\Gamma_{zz}^{b}\right)\\ C_{13}\left(C_{12}\Gamma_{xx}^{b}+C_{22}\Gamma_{yx}^{b}+C_{32}\Gamma_{zx}^{b}\right)+C_{23}\left(C_{12}\Gamma_{xy}^{b}+C_{22}\Gamma_{yy}^{b}+C_{32}\Gamma_{zy}^{b}\right)+C_{33}\left(C_{12}\Gamma_{xz}^{b}+C_{22}\Gamma_{yz}^{b}+C_{32}\Gamma_{zz}^{b}\right)\\ C_{13}\left(C_{13}\Gamma_{xx}^{b}+C_{23}\Gamma_{yx}^{b}+C_{33}\Gamma_{zx}^{b}\right)+C_{23}\left(C_{13}\Gamma_{xy}^{b}+C_{23}\Gamma_{yy}^{b}+C_{33}\Gamma_{zy}^{b}\right)+C_{33}\left(C_{13}\Gamma_{xz}^{b}+C_{23}\Gamma_{yz}^{b}+C_{33}\Gamma_{zz}^{b}\right) \end{array}\right]\\ & =\left[\begin{array}{cccccc} C_{11}^{2} & 2C_{11}C_{21} & 2C_{11}C_{31} & C_{21}^{2} & 2C_{21}C_{31} & C_{31}^{2}\\ C_{11}C_{12} & C_{21}C_{12}+C_{22}C_{11} & C_{31}C_{12}+C_{32}C_{11} & C_{21}C_{22} & C_{31}C_{22}+C_{32}C_{21} & C_{31}C_{32}\\ C_{11}C_{13} & C_{21}C_{13}+C_{23}C_{11} & C_{31}C_{13}+C_{33}C_{11} & C_{21}C_{23} & C_{31}C_{23}+C_{33}C_{21} & C_{31}C_{33}\\ C_{12}^{2} & 2C_{22}C_{12} & 2C_{32}C_{12} & C_{22}^{2} & 2C_{32}C_{22} & C_{32}^{2}\\ C_{12}C_{13} & C_{22}C_{13}+C_{23}C_{12} & C_{32}C_{13}+C_{33}C_{12} & C_{22}C_{23} & C_{32}C_{23}+C_{33}C_{22} & C_{32}C_{33}\\ C_{13}^{2} & 2C_{23}C_{13} & 2C_{33}C_{13} & C_{23}^{2} & 2C_{33}C_{23} & C_{33}^{2} \end{array}\right]\left[\begin{array}{c} \Gamma_{xx}^{b}\\ \Gamma_{xy}^{b}\\ \Gamma_{xz}^{b}\\ \Gamma_{yy}^{b}\\ \Gamma_{yz}^{b}\\ \Gamma_{zz}^{b} \end{array}\right] \end{array}$$

Therefore, the gravity gradient vector matrix $A_{b}^{g}$ is:$$A_{b}^{g}=\left[\begin{array}{cccccc} C_{11}^{2} & 2C_{11}C_{21} & 2C_{11}C_{31} & C_{21}^{2} & 2C_{21}C_{31} & C_{31}^{2}\\ C_{11}C_{12} & C_{21}C_{12}+C_{22}C_{11} & C_{31}C_{12}+C_{32}C_{11} & C_{21}C_{22} & C_{31}C_{22}+C_{32}C_{21} & C_{31}C_{32}\\ C_{11}C_{13} & C_{21}C_{13}+C_{23}C_{11} & C_{31}C_{13}+C_{33}C_{11} & C_{21}C_{23} & C_{31}C_{23}+C_{33}C_{21} & C_{31}C_{33}\\ C_{12}^{2} & 2C_{22}C_{12} & 2C_{32}C_{12} & C_{22}^{2} & 2C_{32}C_{22} & C_{32}^{2}\\ C_{12}C_{13} & C_{22}C_{13}+C_{23}C_{12} & C_{32}C_{13}+C_{33}C_{12} & C_{22}C_{23} & C_{32}C_{23}+C_{33}C_{22} & C_{32}C_{33}\\ C_{13}^{2} & 2C_{23}C_{13} & 2C_{33}C_{13} & C_{23}^{2} & 2C_{33}C_{23} & C_{33}^{2} \end{array}\right].$$
where $C_{ij}$ is the element of the *i*th row and the *j*th column of $C_{g}^{b}$. For the transformation matrix $C_{g}^{b}$ is related to the attitude of the carrier, the gravity gradient vector matrix $A_{b}^{g}$ is also related to the attitude of the carrier. Thus, if you only need to know the attitude of the carrier, you can get the gravity gradient tensor of the FATG in the g-frame.

In order to facilitate the simulation analysis, the center gradient of FTAGG is used to represent the measurement gradient of FTAGG. Assuming that the detected object is a homogeneous cuboid, let its centroid coordinate is Q(W,D,H), its mass density be ρ, its width, height, depth are w,d,h, respectively. Let any coordinate in the cuboid body is P(x,y,z). A schematic diagram of the cuboid acting on the FTAGG is shown in Figure 1.

In the b-frame, the expression of the center gradient caused by the cuboid for FTAGG is: (8)$$\left\{ \begin{array}{l} \Gamma_{xx}^{b}=G\rho \int_{W-\frac{w}{2}}^{W+\frac{w}{2}}dx\int_{D-\frac{d}{2}}^{D+\frac{d}{2}}dy\int_{H-\frac{h}{2}}^{H+\frac{h}{2}}\left[\frac{3x^{2}}{{\left(x^{2}+y^{2}+z^{2}\right)}^{5/2}}-\frac{1}{{\left(x^{2}+y^{2}+z^{2}\right)}^{3/2}}\right]dz\\ \Gamma_{yy}^{b}=G\rho \int_{W-\frac{w}{2}}^{W+\frac{w}{2}}dx\int_{D-\frac{d}{2}}^{D+\frac{d}{2}}dy\int_{H-\frac{h}{2}}^{H+\frac{h}{2}}\left[\frac{3y^{2}}{{\left(x^{2}+y^{2}+z^{2}\right)}^{5/2}}-\frac{1}{{\left(x^{2}+y^{2}+z^{2}\right)}^{3/2}}\right]dz\\ \Gamma_{zz}^{b}=G\rho \int_{W-\frac{w}{2}}^{W+\frac{w}{2}}dx\int_{D-\frac{d}{2}}^{D+\frac{d}{2}}dy\int_{H-\frac{h}{2}}^{H+\frac{h}{2}}\left[\frac{3z^{2}}{{\left(x^{2}+y^{2}+z^{2}\right)}^{5/2}}-\frac{1}{{\left(x^{2}+y^{2}+z^{2}\right)}^{3/2}}\right]dz\\ \Gamma_{xy}^{b}=\Gamma_{yx}^{b}=G\rho \int_{W-\frac{w}{2}}^{W+\frac{w}{2}}dx\int_{D-\frac{d}{2}}^{D+\frac{d}{2}}dy\int_{H-\frac{h}{2}}^{H+\frac{h}{2}}\frac{xy}{{\left(x^{2}+y^{2}+z^{2}\right)}^{5/2}}dz\\ \Gamma_{xz}^{b}=\Gamma_{zx}^{b}=G\rho \int_{W-\frac{w}{2}}^{W+\frac{w}{2}}dx\int_{D-\frac{d}{2}}^{D+\frac{d}{2}}dy\int_{H-\frac{h}{2}}^{H+\frac{h}{2}}\frac{xz}{{\left(x^{2}+y^{2}+z^{2}\right)}^{5/2}}dz\\ \Gamma_{yz}^{b}=\Gamma_{zy}^{b}=G\rho \int_{W-\frac{w}{2}}^{W+\frac{w}{2}}dx\int_{D-\frac{d}{2}}^{D+\frac{d}{2}}dy\int_{H-\frac{h}{2}}^{H+\frac{h}{2}}\frac{yz}{{\left(x^{2}+y^{2}+z^{2}\right)}^{5/2}}dz \end{array}\right.$$
where *G* is Newton’s gravitational constant. Substituting Equation (Equation 8) into Equation (Equation 7), will yield the gravity gradient tensor in g-frame. Assume that the attitude angle of the carrier are ψ, λ, φ, the gravity gradient tensor error for the before and after the change in the attitude of the carrier is: (9)$$\begin{array}{c} \begin{array}{c} \begin{array}{cc} \left[\begin{array}{c} \Delta \Gamma_{xx}^{g}\left(\psi ,\lambda ,\varphi \right)\\ \Delta \Gamma_{xy}^{g}\left(\psi ,\lambda ,\varphi \right)\\ \Delta \Gamma_{xz}^{g}\left(\psi ,\lambda ,\varphi \right)\\ \Delta \Gamma_{yy}^{g}\left(\psi ,\lambda ,\varphi \right)\\ \Delta \Gamma_{yz}^{g}\left(\psi ,\lambda ,\varphi \right)\\ \Delta \Gamma_{zz}^{g}\left(\psi ,\lambda ,\varphi \right) \end{array}\right] & =T_{b}-T_{g}=\left(I-A_{b}^{g}\right)\left[\begin{array}{c} \Gamma_{xx}^{b}\left(\psi ,\lambda ,\varphi \right)\\ \Gamma_{xy}^{b}\left(\psi ,\lambda ,\varphi \right)\\ \Gamma_{xz}^{b}\left(\psi ,\lambda ,\varphi \right)\\ \Gamma_{yy}^{b}\left(\psi ,\lambda ,\varphi \right)\\ \Gamma_{yz}^{b}\left(\psi ,\lambda ,\varphi \right)\\ \Gamma_{zz}^{b}\left(\psi ,\lambda ,\varphi \right) \end{array}\right] \end{array}. \end{array} \end{array}$$
where I is unitmatrix of order 6. $\Delta \Gamma_{ij}^{g}$ is self-gradient of FTAGG, in other words, it is also gravity gradiometry error. It contains carrier attitude information, so it is a function of the attitude of the carrier. The measurement of the self-gradient caused by the attitude of the carrier is generally carried out on the ground. By changing the attitude of the carrier and recording the attitude information and the corresponding self-gradient values, the functional relationship between carrier attitude and self-gradient is established, according to which self-gradient compensation is realized.

In addition to the attitude of the carrier, the fuel mass of carrier also affects gravity gradiometry, the self-gradient caused by the fuel mass is related to the position relative to FTAGG and time, so the influence of fuel mass on the self-gradient is complex. In order to facilitate analysis, suppose the realation between the fuel mass and the fuel consumption time is linear, the self-gradient compensation can be realized according to the relation and the self-gradient corresponding to the attitude of the carrier. Suppose the fuel consumption time is $T_{0}$, at *t* moment, the self-gradient caused by fuel consumption when the attitude angle of the carrier is ψ, λ, φ is: (10)$$\begin{array}{c} \begin{array}{c} \begin{array}{cc} \left[\begin{array}{c} \Delta {\tilde{\Gamma }}_{xx}^{g}\left(t,\psi ,\lambda ,\varphi \right)\\ \Delta {\tilde{\Gamma }}_{xy}^{g}\left(t,\psi ,\lambda ,\varphi \right)\\ \Delta {\tilde{\Gamma }}_{xz}^{g}\left(t,\psi ,\lambda ,\varphi \right)\\ \Delta {\tilde{\Gamma }}_{yy}^{g}\left(t,\psi ,\lambda ,\varphi \right)\\ \Delta {\tilde{\Gamma }}_{yz}^{g}\left(t,\psi ,\lambda ,\varphi \right)\\ \Delta {\tilde{\Gamma }}_{zz}^{g}\left(t,\psi ,\lambda ,\varphi \right) \end{array}\right] & =\left(1-\frac{t}{T_{0}}\right)A_{b}^{g}\left[\begin{array}{c} \Delta \Gamma_{xx}^{b}\left(\psi ,\lambda ,\varphi \right)\\ \Delta \Gamma_{xy}^{b}\left(\psi ,\lambda ,\varphi \right)\\ \Delta \Gamma_{xz}^{b}\left(\psi ,\lambda ,\varphi \right)\\ \Delta \Gamma_{yy}^{b}\left(\psi ,\lambda ,\varphi \right)\\ \Delta \Gamma_{yz}^{b}\left(\psi ,\lambda ,\varphi \right)\\ \Delta \Gamma_{zz}^{b}\left(\psi ,\lambda ,\varphi \right) \end{array}\right] \end{array}. \end{array} \end{array}$$
where $\Delta \Gamma_{ij}^{b}\left(\psi ,\lambda ,\varphi \right)$ is the self-gradient $\Delta \Gamma_{ij}$ caused by fuel mass when the vehicle attitude angle is ψ, λ, φ and there is no fuel consumption in b-frame. In airborne gravity gradiometry, it is necessary to record the attitude and flight time of the carrier in real time. Assume that the attitude angle of the carrier is ψ, λ, φ, at *t* moment, the true value of gravity gradiometry can be calculated using the following formula:(11)$$\begin{array}{c} \begin{array}{c} \left[\begin{array}{c} \Gamma_{xx}^{g}\left(t,\psi ,\lambda ,\varphi \right)\\ \Gamma_{xy}^{g}\left(t,\psi ,\lambda ,\varphi \right)\\ \Gamma_{xz}^{g}\left(t,\psi ,\lambda ,\varphi \right)\\ \Gamma_{yy}^{g}\left(t,\psi ,\lambda ,\varphi \right)\\ \Gamma_{yz}^{g}\left(t,\psi ,\lambda ,\varphi \right)\\ \Gamma_{zz}^{g}\left(t,\psi ,\lambda ,\varphi \right) \end{array}\right]=A_{b}^{g}\left[\begin{array}{c} \Gamma_{xx}^{b}\left(t,\psi ,\lambda ,\varphi \right)\\ \Gamma_{xy}^{b}\left(t,\psi ,\lambda ,\varphi \right)\\ \Gamma_{xz}^{b}\left(t,\psi ,\lambda ,\varphi \right)\\ \Gamma_{yy}^{b}\left(t,\psi ,\lambda ,\varphi \right)\\ \Gamma_{yz}^{b}\left(t,\psi ,\lambda ,\varphi \right)\\ \Gamma_{zz}^{b}\left(t,\psi ,\lambda ,\varphi \right) \end{array}\right]-\left[\begin{array}{c} \Delta \Gamma_{xx}^{g}\left(\psi ,\lambda ,\varphi \right)\\ \Delta \Gamma_{xy}^{g}\left(\psi ,\lambda ,\varphi \right)\\ \Delta \Gamma_{xz}^{g}\left(\psi ,\lambda ,\varphi \right)\\ \Delta \Gamma_{yy}^{g}\left(\psi ,\lambda ,\varphi \right)\\ \Delta \Gamma_{yz}^{g}\left(\psi ,\lambda ,\varphi \right)\\ \Delta \Gamma_{zz}^{g}\left(\psi ,\lambda ,\varphi \right) \end{array}\right]-\left[\begin{array}{c} \Delta {\tilde{\Gamma }}_{xx}^{g}\left(t,\psi ,\lambda ,\varphi \right)\\ \Delta {\tilde{\Gamma }}_{xy}^{g}\left(t,\psi ,\lambda ,\varphi \right)\\ \Delta {\tilde{\Gamma }}_{xz}^{g}\left(t,\psi ,\lambda ,\varphi \right)\\ \Delta {\tilde{\Gamma }}_{yy}^{g}\left(t,\psi ,\lambda ,\varphi \right)\\ \Delta {\tilde{\Gamma }}_{yz}^{g}\left(t,\psi ,\lambda ,\varphi \right)\\ \Delta {\tilde{\Gamma }}_{zz}^{g}\left(t,\psi ,\lambda ,\varphi \right) \end{array}\right] \end{array}. \end{array}$$

The Equtation (Equation 11) is also the expression of the self-gradient compensation, the error of self-gradient can be removed by using the Equation (Equation 11); thus, self-gradient compensation can be realized, and more accurate information of the gravity gradient can be obtained. In the end, the precision of gravity gradient measurement can be significantly improved.

## 3. Self-Gradient Simulation Results

The value of the self-gradient is not only related to the distributed mass of the carrier, but also to the relative position of the distributed mass of the carrier. The structure of the carrier is complex and the fuel mass varies with time and environment. In order to facilitate simulation and analysis, it is assumed that the distribution mass of the carrier wing is symmetrical relative to the center of the carrier. Taking the Cessna208B carrier manufactured by Cessna as the carrier of gravity gradiometer, its parameters are as follows: the fuselage length is 12.7 m, the fuselage height is 4.27 m, the cabin length is 5.4 m, and the cabin width is 1.6 m. The cabin height is 1.3 m, the maximum take-off weight is 3969 kg and the maximum fuel weight is 1019 kg. Considering the influence of fuel on the gravity gradiometry, the fuel is placed in the wing, which can further reduce the influence of the fuel mass change on the gravity gradiometry. The influence of the vehicle attitude and the fuel mass on the gravity gradiometry will be simulated and analyzed in the following.

### 3.1. Self-Gradient Caused by the Attitudes of the Carrier

Assume that the distribution mass of the carrier is divided into five parts, namely: the front part, the middle part, the rear part, the left wing and the right wing. Schematic diagram of the distribution mass of the carrier is shown in Figure 2. The total mass of the carrier is 3000 kg. Since the fuel mass is a function of time, the fuel mass is now out of consideration, the self-gradient caused by fuel mass will be analyzed in Section 3.2. The relevant parameters of the distribution mass of the carrier are set as shown in Table 1.

The range of attitude angle of vehicle is set to 0∼180${}^{\circ }$ and the change of attitude angle is set to $0.001^{\circ }$. According to the Table 1, the self-gradient simulation is carried out under these parameters, and the values of the self-gradient element caused by the carrier in different attitudes are shown in Figure 3, where the value of several element in the five self-gradient elements caused by the distribution mass of the carrier in three attitudes is high. As shown in Figure 3, the attitude of the carrier has the greatest influence on the self-gradient element $\Gamma_{yz}$, and its value exceeds 200 Eu. At the same time, the self-gradient element $\Gamma_{zz}$ in heading attitude, the self-gradient element $\Gamma_{xx}$, $\Gamma_{xy}$ and $\Gamma_{xz}$ in pitch attitude, and the self-gradient element $\Gamma_{yy}$ in roll attitude are zero. These results are helpul to provide engineering guidance for us during the airborne gravity gradiometry. The self-gradient tensor caused by the mass of the carrier in heading is shown in Figure 4. As shown in Figure 4, the self-gradient tensor element $\Gamma_{xx},\Gamma_{xy},\Gamma_{xz},\Gamma_{yy}$ and $\Gamma_{yz}$ are greatly affected by the middle part of the carrier in heading attitude, and the self-gradient element $\Gamma_{zz}$ caused by all the distribution mass of the carrier in heading attitude is about zero. Self-gradient tensor caused by the mass of the carrier in pitch is shown in Figure 5. As shown in Figure 5, the self-gradient tensor element $\Gamma_{yy},\Gamma_{yz}$ and $\Gamma_{zz}$ are greatly affected by the middle part of the carrier, and the $\Gamma_{xy}$ and $\Gamma_{xz}$ are greatly affected by the wing of the carrier, and the self-gradient element $\Gamma_{xx}$ caused by all the distribution mass of the carrier in pitch attitude is about zero. The self-gradient tensor caused by the mass of the carrier in roll is shown in Figure 6. As shown in Figure 6, the self-gradient tensor element $\Gamma_{xx},\Gamma_{xy},\Gamma_{xz},\Gamma_{yz}$ and $\Gamma_{zz}$ are greatly affected by the middle part of the carrier, and the self-gradient element $\Gamma_{yy}$ caused by all the distribution mass of the carrier in roll attitude is about zero. Another important conclusion is that the self-gradient tensor caused by the wing of the carrier is small in the Figure 4, Figure 5 and Figure 6.

By changing the attitude of the carrier, the corresponding relationship between the attitude of the carrier and the self-gradient caused by the carrier can be established. The relation between the self-gradient and the attitude of the carrier can be approximated by polynomial function, or a database can be established to record the self-gradient value corresponding to each attitude angle of the carrier. During the airborne gravity gradiometry, the carrier attitude is measured by the platform angle sensor, the corresponding values of the plane attitude angle and the self-gradient caused by the carrier are calculated, and then the self-gradients are removed from the gravity gradient information. Thus, the gravity gradient compensation is realized.

### 3.2. Self-Gradient Caused by the Fuel Mass of the Carrier

As the fuel continues to be consumed during the exploration, the fuel mass is getting smaller and smaller. Because the fuel mass varies over time, the self-gradient caused by the fuel is related not only to the attitude of the carrier, but also to the time. Therefore, it can be concluded that the self-gradient caused by the fuel mass is a function related to the attitude and flight time of the carrier. In fact, the fuel mass is not a constant quantity, it will change with the altitude, pressure, temperature and humidity of the carrier, and the change of the flight environment will also lead to the atmospheric pressure near the FTAGG.

The influence of temperature and other factors on the fuel mass will be more complicated. In order to achieve the high precision of the self-gradient compensation, the fuel mass needs real-time monitoring. In order to facilitate analysis, it is assumed that the fuel mass is a linear relationship with time, and the flight time for the carrier is set to 6 hour. The fuel density is set to 780 $\mathrm{kg}/m^{3}$, and the initial fuel volume is set to 1.8 m×2.8 m×0.6 m, and the self-gradient simulation is carried out under this parameters. The self-gradient caused by the fuel mass of the carrier with time in heading are shown in Figure 7, self-gradient tensor element $\Gamma_{xx},\Gamma_{xy},\Gamma_{xz},\Gamma_{yy}$ and $\Gamma_{yz}$ are greatly affected by the fuel mass of the carrier. The self-gradient caused by the fuel mass of the carrier with time in pitch are shown in Figure 8, Self-gradient tensor element $\Gamma_{yy},\Gamma_{yz}$ and $\Gamma_{zz}$ are greatly affected by the fuel mass of the carrier. The self-gradient caused by the fuel mass of the carrier with time in roll are shown in Figure 9, Self-gradient tensor element $\Gamma_{xx},\Gamma_{xy},\Gamma_{xz},\Gamma_{yz}$ and $\Gamma_{zz}$ are greatly affected by the fuel mass of the carrier. As shown in Figure 7, Figure 8 and Figure 9, Self-gradient tensor element $\Gamma_{yz}$ is affected by the fuel mass of the carrier in three attitudes. The self-gradient caused by fuel has seriously affected the precision of gravity gradiometry, in the analysis of the self-gradient caused by the fuel, real-time monitoring of fuel mass is required. In the process of computing the self-gradient, it is necessary to consider the attitude of the aircraft and the fuel quality to make the self-gradient compensation in real time. In this paper, in order to better express the effects of aircraft attitude and fuel consumption on the self-gradient, using the cuboid instead of fuel is an approximate representation of the mass distribution of the carrier, which provides a guidance for airborne gravity gradient compensation.

## 4. Simulation of Self-Gradient Compensation

A simulation experiment is developed based on the above analysis. In order to verify the feasibility of the self-gradient compensation method, a five-stage flight state is used to simulate the self-gradient of the distribution mass and fuel mass for the carrier, and then the self-gradient compensation is computed based on Equation (Equation 11). Assuming that the flight time is 6 h, the flight altitude is 100 m, the sampling interval of gravity gradient information space is 50 m, the flight attitude parameters of the carrier during the five flight stages are shown in Table 2. Assuming that the gravity gradient anomalies are given rise by the subsurface body(anomaly-body), in order to simulate and describe the gravity gradient anomaly information, the subsurface body can be considered to be a cuboid, and the density contrast and volume of the cuboid are set to 2000 $\mathrm{kg}/m^{3}$ and 20 km× 1 km× 1 km, respectively. The center coordinate of gravity gradient anomaly body is (0,−2,−1.1) km, and the starting and terminating coordinates of the vehicle are (−60,0,0) km and (60,0,0) km, respectively. The real ideal gravity gradient value of the anomaly-body is shown in Figure 10a. As can be seen in Figure 10a, there are two sharps of the gravity gradient anomalies response in about 50 km and 70 km, where the reason is that the position of the sharp response point is the edge of the anomaly-body. Before self-gradient compensation, the simulation result of gravity gradiometry for anomaly-body is shown in Figure 10b. As can be seen in Figure 10b, the gravity gradient elements of the anomaly-body are seriously disturbed, which means it is difficult to discern the exact gravity gradient values. After Self-gradient compensation, gravity gradient value of the anomaly-body is shown in Figure 10c. The interference gravity gradient data is removed (this can be observed in Figure 10c). The error of self-gradient compensation is shown in Figure 11. The self-gradient compensation error is within 0.1 Eu, the self-gradient compensation method that we present can meet the requirement of high-precision gravity gradiometry.

## 5. Conclusions

Because FTAGG is extremely sensitive to its operating environment, the attitude of the carrier and the fuel mass will seriously affect the accuracy of gravity gradiometry for the FTAGG, which is analyzed and studied here. In this paper, a simulation experiment on the self-gradient compensation method is carried out, the error of the self-gradient compensation is within 0.1 Eu. The proposed self-gradient compensation method is verified. The self-gradient compensation method we present can improve the precision of gravity gradiometry. More importantly, it might provide a new and potential ideal for the self-gradient compensation in airborne gravity gradiometry. Nevertheless, this proposed method still has some flaws and expected challenges. For example, first, the precision of attitude measurement of vehicle will have a certain degree of influence on the self-gradient compensation. Second, it is difficult to determine the shape and the distributed mass of the carrier because of the complexity of the mass distribution inside the carrier, so these will have a great influence on the self-gradient compensation. Third, because the fuel mass of the carrier is easily affected by the external environment, the fuel shape and mass of the carrier cannot be accurately measured, so this is also an important factor affecting the accuracy of self-gradient compensation. 

## Figures and Tables

**Figure 1 sensors-19-01950-f001:**
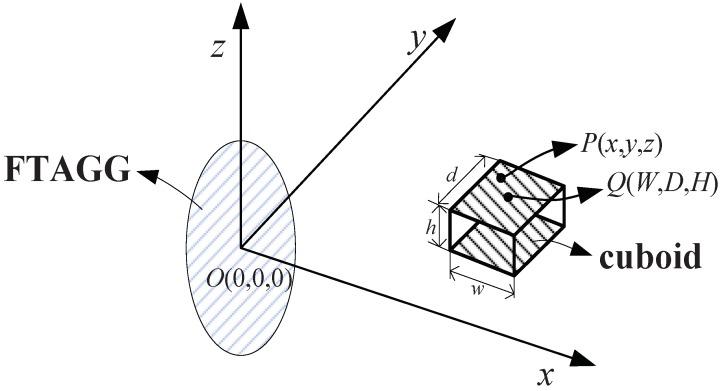
Schematic diagram of the cuboid acting on the full-tensor airborne gravity gradiometer (FTAGG).

**Figure 2 sensors-19-01950-f002:**
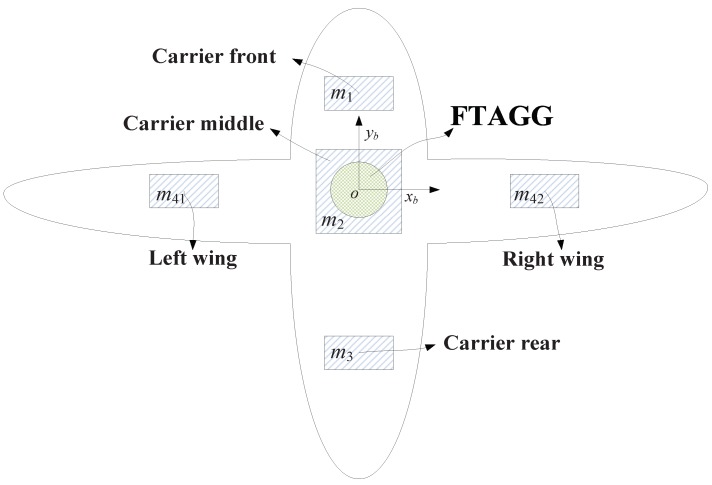
Schematic diagram of the distribution mass for the carrier.

**Figure 3 sensors-19-01950-f003:**
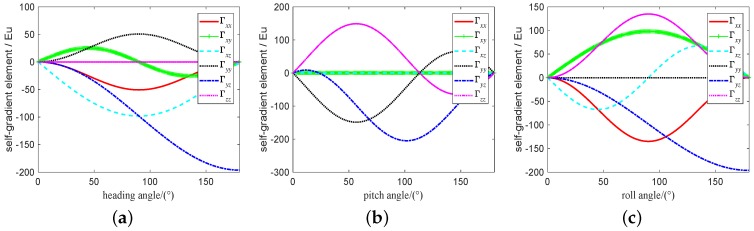
Self-gradient caused by the distribution mass of the carrier in different attitudes. (**a**) Self-gradient caused by the distribution mass of the carrier in heading attitude. (**b**) Self-gradient caused by the distribution mass of the carrier in pitch attitude. (**c**) Self-gradient caused by the distribution mass of the carrier in roll attitude.

**Figure 4 sensors-19-01950-f004:**
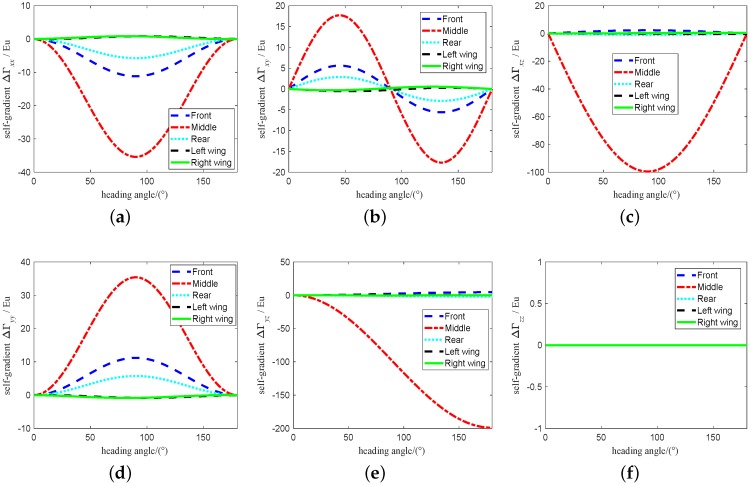
Self-gradient caused by the distribution mass of the carrier in heading attitude. (**a**) Self-gradient element $\Gamma_{xx}$. (**b**) Self-gradient element $\Gamma_{xy}$. (**c**) Self-gradient element $\Gamma_{xz}$. (**d**) Self-gradient element $\Gamma_{yy}$. (**e**) Self-gradient element $\Gamma_{yz}$. (**f**) Self-gradient element $\Gamma_{zz}$.

**Figure 5 sensors-19-01950-f005:**
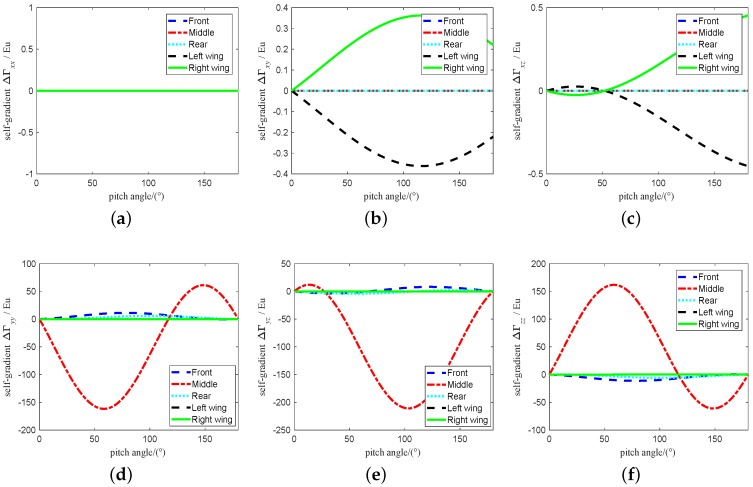
Self-gradient caused by the distribution mass of the carrier in pitch attitude. (**a**) Self-gradient element $\Gamma_{xx}$. (**b**) Self-gradient element $\Gamma_{xy}$. (**c**) Self-gradient element $\Gamma_{xz}$. (**d**) Self-gradient element $\Gamma_{yy}$. (**e**) Self-gradient element $\Gamma_{yz}$. (**f**) Self-gradient element $\Gamma_{zz}$.

**Figure 6 sensors-19-01950-f006:**
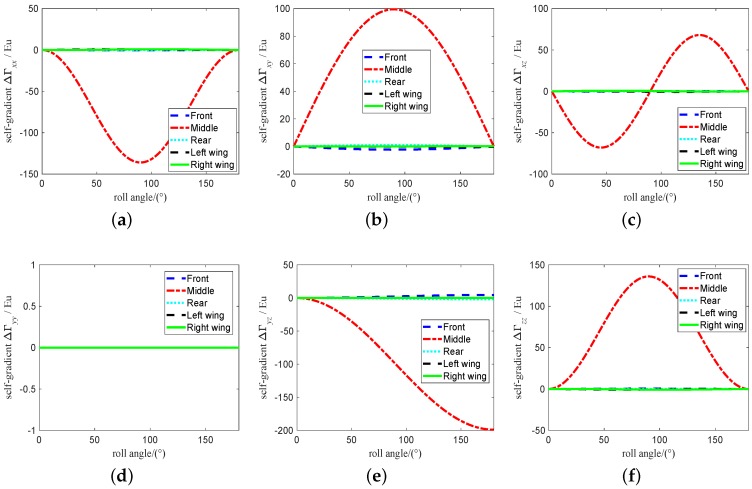
Self-gradient caused by the distribution mass of the carrier in roll attitude. (**a**) Self-gradient element $\Gamma_{xx}$. (**b**) Self-gradient element $\Gamma_{xy}$. (**c**) Self-gradient element $\Gamma_{xz}$. (**d**) Self-gradient element $\Gamma_{yy}$. (**e**) Self-gradient element $\Gamma_{yz}$. (**f**) Self-gradient element $\Gamma_{zz}$.

**Figure 7 sensors-19-01950-f007:**
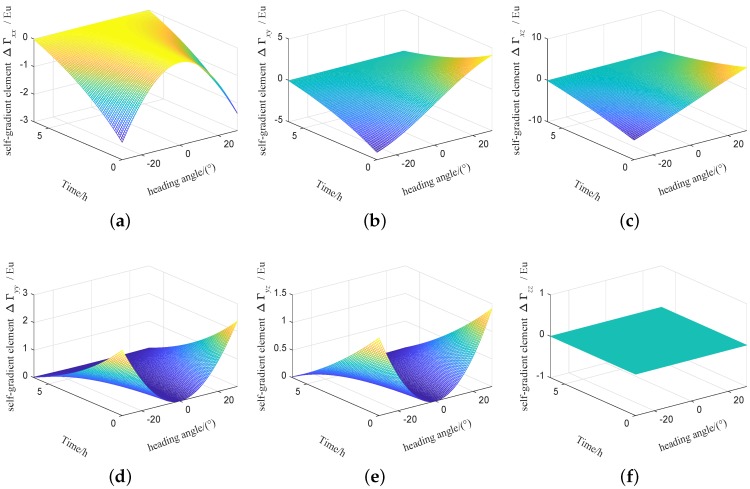
Self-gradient caused by the fuel mass of the carrier in heading attitude. (**a**) Self-gradient element $\Gamma_{xx}$. (**b**) Self-gradient element $\Gamma_{xy}$. (**c**) Self-gradient element $\Gamma_{xz}$. (**d**) Self-gradient element $\Gamma_{yy}$. (**e**) Self-gradient element $\Gamma_{yz}$. (**f**) Self-gradient element $\Gamma_{zz}$.

**Figure 8 sensors-19-01950-f008:**
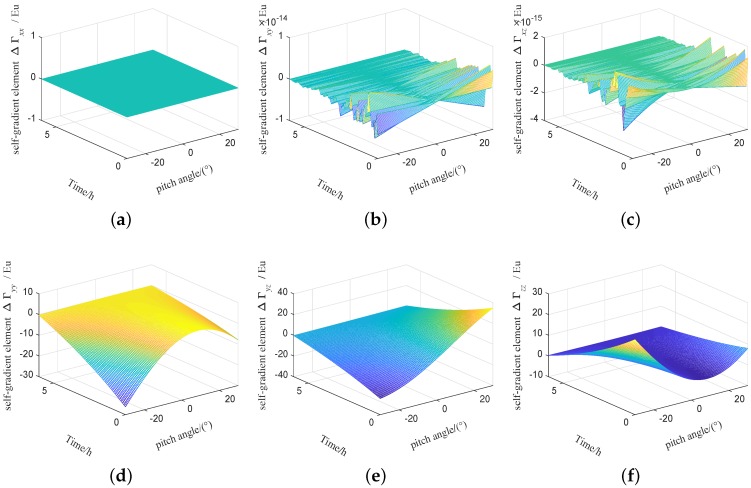
Self-gradient caused by the fuel mass of the carrier in pitch attitude. (**a**) Self-gradient element $\Gamma_{xx}$. (**b**) Self-gradient element $\Gamma_{xy}$. (**c**) Self-gradient element $\Gamma_{xz}$. (**d**) Self-gradient element $\Gamma_{yy}$. (**e**) Self-gradient element $\Gamma_{yz}$. (**f**) Self-gradient element $\Gamma_{zz}$.

**Figure 9 sensors-19-01950-f009:**
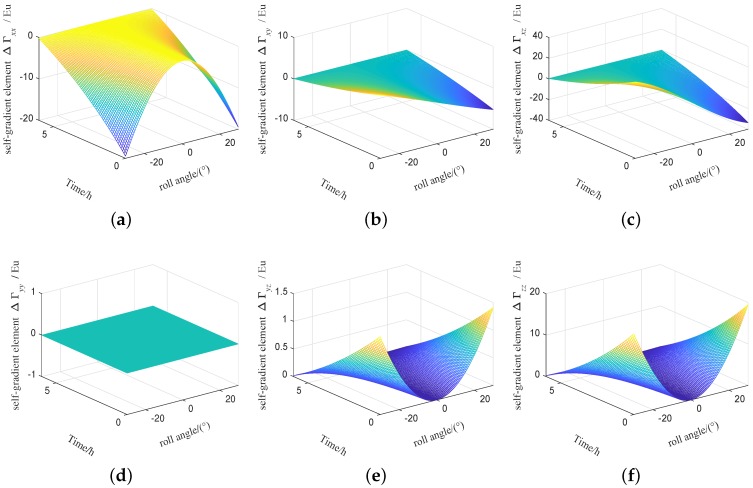
Self-gradient caused by the fuel mass of the carrier in roll attitude. (**a**) Self-gradient element $\Gamma_{xx}$. (**b**) Self-gradient element $\Gamma_{xy}$. (**c**) Self-gradient element $\Gamma_{xz}$. (**d**) Self-gradient element $\Gamma_{yy}$. (**e**) Self-gradient element $\Gamma_{yz}$. (**f**) Self-gradient element $\Gamma_{zz}$.

**Figure 10 sensors-19-01950-f010:**
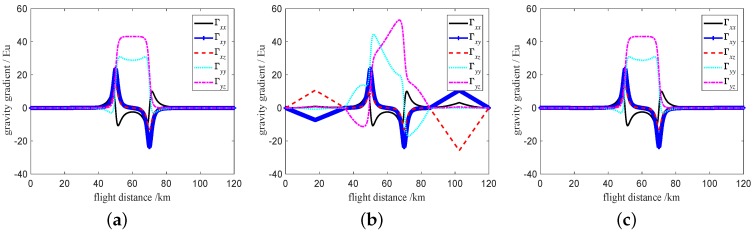
Gravity gradient real ideal value and results of the self-gradient compensation. (**a**) Real ideal gravity gradient value of the anomaly-body. (**b**) Before self-gradient compensation. (**c**) After Self-gradient compensation.

**Figure 11 sensors-19-01950-f011:**
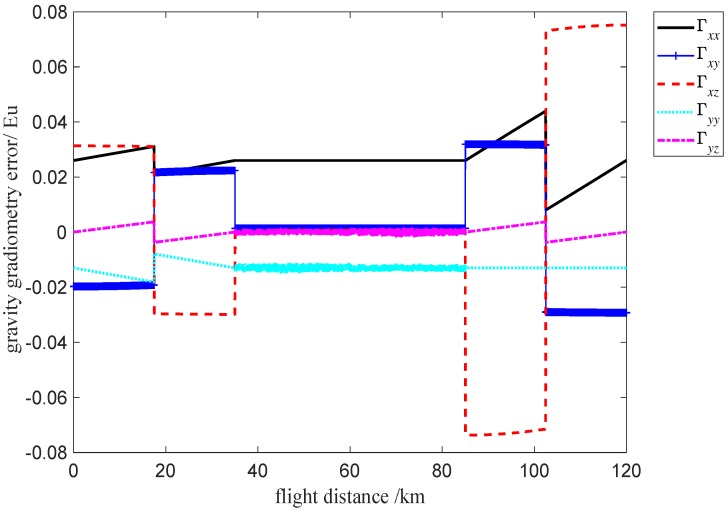
Self-gradient compensation error.

**Table 1 sensors-19-01950-t001:** Parameters of the distribution mass for the carrier.

Distribution Mass Name	Volume (m×m×m)	Density ($\mathrm{kg}/m^{3}$)	Position {(x,y,z)m}
Front ($m_{1}$)	1×1×0.8	1200	(0, 2.5, 0.5)
Middle ($m_{2}$)	1×1×0.5	1000	(0, 0.5, 0.5)
Rear ($m_{3}$)	1.8×1.2×0.5	800	(0, −3.0, 0.5)
Left wing ($m_{41}$)	2×1×0.3	500	(−4.0, 0.5, 1.0)
Right wing ($m_{42}$)	2×1×0.3	500	(4.0, 0.5, 1.0)

**Table 2 sensors-19-01950-t002:** Parameters of flight attitude for the carrier.

Information of Flight	Attitude of Carrier
0∼17.5 km	$\psi =0.02N^{\circ }\left(N=0,1,2,...,350\right)$
17.5 km∼35 km	$\psi =7^{\circ }-0.02N^{\circ }\left(N=0,1,2,...,350\right)$
35 km∼85 km	$\lambda =5\sin{\left(0.002\pi N\right)}^{\circ }\left(N=0,1,2,...,1000\right)$
85 km∼102.5 km	$\varphi =-0.02N^{\circ }\left(N=0,1,2,...,350\right)$
102.5 km∼120 km	$\varphi =-7^{\circ }+0.02N^{\circ }\left(N=0,1,2,...,350\right)$

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
