# Peer review of "Self-Gradient Compensation of Full-Tensor Airborne Gravity Gradiometer"

_sensors, 2019, doi:10.3390/s19081950_

Reviewer 1 Report

Dear authors,

it is a principally interesting contribution but you present too many equations that

are standard but on the other hand you fail to provide numerical values (before the results section) so the presentation is sort of hard to swallow.

In the results section you simply present diagrams without a suficient discusion. You can't (in my opinion) leave it to the reader to make sense out of the presentation.

I do attach a commented file for your reference.

Best wishes

Author Response

Thank you for your careful review. I have made the following modifications according to your request and attached the paper.

page1,2: the maning is: the distribution mass of the carrier is in the middle of the carrier

page2,2: yes

page5,1: these equations are written to show how to realize self-gradient compensation, so I think these equations do help to this paper.

2: yes, I can, but need many data, these is not many space to write all the data in the paper.

page7,2:not need

page10,12,1:ok,Additions have been made.

Reviewer 2 Report

The paper describes the effect on the component of the gravity second derivatives due to the variation of the masses geometry of an aircraft and of the variation of fuel masses, when performing gravity gradiometry.

There are several typos and small errors, and a few sentences that I had trouble following. I think detailed copyediting would be helpful.

I guess that the correct symbol for Eotvos is just a capital E. So please correct it.

I suggest to simplify Section two. For instance Eq.5, 8 and 9 can be, in my opinion, safely removed.

Considering Eq. 10, how does the computation of the gravity gradients is perfomed? I guess it is done by Nagy formulas for the right rectangular prism. In case, please state it explicitly.

In order figures readability please increase the size of font in Figures from 3 to 10.

Author Response

Thank you for your careful review. I have made the following modifications according to your request and attached the paper. 

about the unit of gravity gradient, up to now, the unit of gravity gradient has not been formally determined yet, so the unit of gravity gradient can be E or Eu or the other symbol.

Round  2

Reviewer 1 Report

Dear authors, thank you very much for the corrections, I think, it's fine now.